# Optimizing the internal phase reference to shape the output of a multimode optical fiber

Liam Collard[1,2]*, Linda Piscopo[1,3], Filippo Pisano[1,4], Di Zheng[1], Massimo De Vittorio[1,2,3‡]*, Ferruccio Pisanello[1,2‡]*

1 Istituto Italiano di Tecnologia, Center for Biomolecular Nanotechnologies, Arnesano, Italy, 2 RAISE Ecosystem, Genova, Italy, 3 Dipartimento di Ingegneria Dell'Innovazione, Università del Salento, Lecce, Italy, 4 Department of Physics and Astronomy "G. Galilei", University of Padova, Padova, Italy

‡ MDV and FP jointly supervised and are co-last authors of this work.
* liam.collard@iit.it (LC); massimo.devittorio@iit.it (MDV); ferruccio.pisanello@iit.it (FP)

**Data Availability Statement:** All relevant data are within the manuscript and its Supporting Information files.

## Abstract

Pre-shaping light to achieve desired amplitude distributions at the tip of a multimode fiber (MMF) has emerged as a powerful method allowing a wide range of imaging techniques to be implemented at the distal facet. Such techniques rely on measuring the transmission matrix of the optically turbid waveguide which scrambles the coherent input light into an effectively random speckle pattern. Typically, this is done by measuring the interferogram between the output speckle and a reference beam. In recent years, an optical setup where the reference beam passes through the MMF has become an attractive configuration because of the high interferometric stability of the common optical path. However, the merits and drawbacks of an internal reference beam remain controversial. The measurement of the transmission matrix is known to depend on the choice of internal reference and has been reported to result in "blind spots" due to phase singularities of the reference beam. Here, we describe how the focussing efficiency of the calibration can be increased by several percent by optimising the choice of internal reference beam.

## Introduction

Wavefront shaping has emerged as a powerful method to control the transmission through a multimode fiber (MMF) allowing the full modal diversity of the waveguide to be exploited [1–9] By pre-shaping the light prior to transmission, reconfigurable focussed spots can be generated at the distal facet of the fiber. This has enabled a plethora of imaging techniques at the tip of the fiber including in-vivo fluorescence [10, 11], Raman [12, 13], CARS [14], second harmonic generation [15] and also more complex photonic techniques such as optical tweezers [16], activation of plasmonic nanostructures [17] and also through multicore fibers [18–22]. Recently, a new generation of MMF based endoscopes capable of imaging objects distal from the fiber have been developed by shaping the output light in the Fourier plane [23, 24].

All these techniques are reliant on the ability to accurately measure the transmission matrix of the waveguide. Typically, this is done by splitting the beam prior to transmission through the fiber to create an external reference beam, which is re-joined to the beam passing through

**Funding:** M.D.V. and Fe.P. jointly supervised and are co-last authors of this work. L.C., D.Z., M.D.V., and Fe.P. acknowledge funding from the European Union's Horizon 2020 Research and Innovation Program under Grant Agreement No. 828972. L.C., M.D.V. and Fe.P. acknowledge funding from the Project "RAISE (Robotics and AI for Socio-economic Empowerment)" code ECS00000035 funded by European Union – NextGenerationEU PNRR MUR - M4C2 – Investimento 1.5 - Avviso "Ecosistemi dell'Innovazione" CUP J33C22001220001. Fi.P., and Fe.P. acknowledge funding from the European Research Council under the European Union's Horizon 2020 Research and Innovation Program under Grant Agreement No. 677683. Fi.P., M.D.V., and Fe.P. acknowledge funding from the European Union's Horizon 2020 Research and Innovation Program under Grant Agreement No 101016787. M.D.V. acknowledges funding from the European Research Council under the European Union's Horizon 2020 Research and Innovation Program under Grant Agreement No. 692943. M.D.V. acknowledges funding from the U. S. National Institutes of Health (Grant No. U01NS094190). M.D.V., and Fe.P. acknowledge funding from the U.S. National Institutes of Health (Grant No. 1UF1NS108177-01). M,D,V and Fe,P acknowledge funding from European Research Council under the European Union's Horizon 2020 Research and Innovation Program under Grant Agreement No. 966674

**Competing interests:** I have read the journal's policy and the authors of this manuscript have the following competing interests: M.D.V. and F. Pisanello are founders and hold private equity in OptogeniX srl, a company that develops, produces and sells technologies to deliver light into the brain. This does not alter our adherence to PLOS ONE policies on sharing data and materials. OptogeniX did not fund the research described in this work. M.D.V.: OptogeniX srl (I). F.P.: OptogeniX srl (I).

the fiber after the fiber output, to create an interference pattern. The phase difference between each output mode and the reference can then be measured and re-modulated to generate constructive interference. The result is a set of diffraction limited foci scanning the output plane. The use of external reference beam has achieved considerable success, with one caveat being the need of a highly stable interferometric setup and optical alignment. Therefore, as the field develops and more complex multispectral imaging modalities, simplified transmission matrix measurements methods are of keen interest. In this respect, "*reference free*", phase retrieval techniques [25, 26] and machine learning algorithms [27, 28] have both emerged as elegant alternatives, as they require no kind of interferomic measurement however they generally require larger datasets/more complex computational techniques and are generally less accurate than interferomic measurements

An *internal reference beam* passing through the core of the fiber is an interesting alternative. Such a system is intrinsically reconfigurable, provides high interferometric stability and is a simpler optical setup due to the common optical path. Indeed, recent works on distal imaging endoscopes have employed internal reference beams for such reasons [29]. However, it has been demonstrated that the use of an internal reference beam results in *blind spots* at the output due to vortex singularities in the reference beam and can also lead to an uneven focussing efficiency over the fiber core [29]. A proposed solution to this has been to use multiple internal reference beams, at the price of a greater computational complexity [30]. Thus, the merits and drawbacks of internal reference beams remain controversial. In this work we aim to provide a detailed assessment of both surface and Fourier plane calibrations (where the light is focussed on the distal facet or gets collimated in a beam of low divergence emerging from the distal facet, respectively) performed with single and serialised internal reference beams, striving at optimizing the relative focussing efficiency of the system.

## Experimental methods

### Optical system

The employed wavefront shaping setup is shown in **Fig 1A**. A continuous wave 633 nm beam was expanded by a telescope formed of lenses L1 and L2 to overfill the screen of a spatial light modulator (SLM) (ODP512, Meadowlark optics). Prior to this, the polarisation was rotated by a half wave plate for optimal modulation. The screen of the SLM was conjugated with the back aperture of microscope objective MO1 (0.65 NA, 40x, AMEP4625, ThermoFisher) by a 4f system comprised of lenses L3 and L4. MO1 focused the modulated light onto the proximal facet of the MMF (0.22 NA, 50 μm core diameter, Thorlabs FG050UGA). The numerical aperture of MO1 was significantly higher than that of the MMF. This was done so as the number of pairs $(j_x, j_y)$ should roughly match the number of modes carried by the fiber [2]. The number of modes supported by the fiber is given by $M = \frac{V^2}{2}$ where $V = \frac{2\pi r}{\lambda} NA = \frac{2 \times \pi \times 25}{0.633} \times 0.22 = 54.59$. Thus, $M \approx 1490$ (or 745 per polarisation). Therefore, the minimum lateral resolution at the input should be $\frac{2r}{\sqrt{M_P}} = \frac{50\mu m}{27.3} = 1.83$ um. Meanwhile, Abbe's diffraction formula for lateral (XY) resolution calculated with MO1 is $d = \frac{\lambda}{2NA} = \frac{.633}{2*0.65} = 0.48$ um. The transmission efficiency from MO1 to the MMF is approximately 30%".

For alignment purposes, the reflection of the input facet was monitored on CCD1. The fiber transmission was collected by MO2 (10x, 0.3 NA, MPLFLN10x—Olympus) and guided onto CCD2 (monitoring the distal fiber facet) and CCD3 (monitoring the transmission in the Fourier plane). The surface of the proximal and distal facets are defined as $(x_{in}, y_{in})$ $(x_{out}, y_{out})$ and the Fourier plane of each facet $(u_{in}, v_{in})$ $(u_{out}, v_{out})$ respectively.

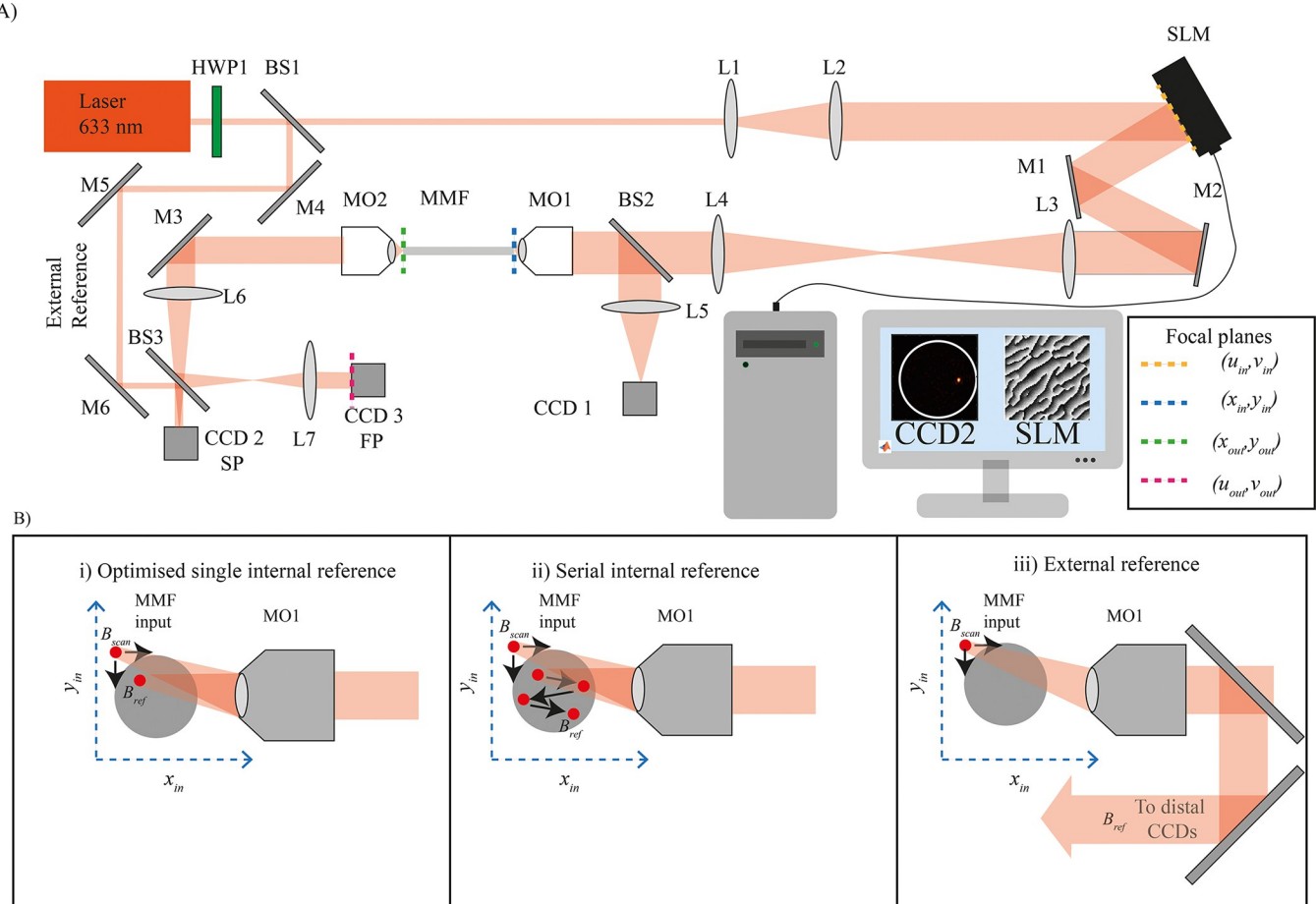

**Fig 1.** A) Optical system for shaping light at the tip of a MMF using an internal reference beam. HWP- half wave plate, L- lens, SLM- spatial light modulator, M- mirror, BS- beam splitter, MO—microscope objective, MMF–multimode fiber, CCD- charged coupling device. The four focal planes $(u_{in}, v_{in})$, $(x_{in}, y_{in})$, $(x_{out}, y_{out})$, $(u_{out}, v_{out})$ correspond to the screen of the SLM, proximal fiber facet, distal fiber surface plane, distal Fourier plane and are indicated by the purple, blue, green and brown dashed lines respectively. B) Illustration of the three calibration techniques applied in this work utilising i) position of single internal reference ii) serialised internal references with optimized positions iii) an external reference.

### Calibrating with single or serial internal references

Prior to the calibration, sawtooth gratings $\phi_{scan}^{j_x, j_y, p}(u_{in}, v_{in}) = mod(au_{in} + bv_{in} + p, 2\pi)$ and $\phi_{ref}(u_{in}, v_{in}) = mod((cu_{in} + dv_{in}), 2\pi)$ were defined, where $a$ and $b$ set the periodicity of the grating along $x_{in}$, and $y_{in}$ so the beam $B_{scan}^{j_x, j_y, p}$ raster scans a $25 \times 25$ grid overlaying the input facet of the MMF indexed by $(j_x, j_y)$. $c$ and $d$ were chosen so that the reference beam $B_{ref}$ was fixed. $p$ represents the phase shift applied to the scanning points incremented in steps of $\frac{\pi}{2}$ (for a total of 4 phase steps). $\phi_{interfere}^{j_x, j_y, p}(u_{in}, v_{in}) = arg(exp(i\phi_{scan}^{j_x, j_y, p}(u_{in}, v_{in})) + exp(i\phi_{ref}))$ could then generate both $B_{scan}^{j_x, j_y, p}$ and $B_{ref}$ simultaneously. Both sets of $\phi_{interfere}^{j_x, j_y, p}$ and $\phi_{scan}^{j_x, j_y, p}$ were saved as.bmp files prior to the calibration and $\phi_{scan}^{j_x, j_y, p}$ was loaded onto the graphics processing unit (GPU—NVIDIA GeForce GTX 960) [23].

The first part of the calibration consisted of measuring the transmission of $B_{interfere}^{j_x, j_y, p}(u_{in}, v_{in})$ through the MMF on the Surface Plane (SP) and Fourier Plane (FP) of the output facet, hereafter referred to as $f_{SP}(B_{interfere}^{j_x, j_y, p})$ and $f_{FP}(B_{interfere}^{j_x, j_y, p})$, for each value $j_x, j_y, p$ at a $30 \times 30$ set of points in both output planes $(x_{out}, y_{out})$, $(u_{out}, v_{out})$ which were measured on CCD2 and CCD 3 respectively. "To

acquire a single speckle pattern during the calibration step, the total exposure time of the camera was 72 ms and the SLM was given 150 ms to refresh between each step to ensure no cross-talk. For each pair $(j_x, j_y)$, the phase shift $p$ generating the maximum intensity of $f_{SP}(B_{interfere}^{j_x,j_y,p})$ or $f_{FP}(B_{interfere}^{j_x,j_y,p})$ at each of the $30 \times 30$ pixels in the $(x_{out}, y_{out})$, $(u_{out}, v_{out})$ planes, denoted, $p_{opt,SP}^{j_x,j_y,(x_{out},y_{out})}$ and $p_{opt,FP}^{j_x,j_y,(u_{out},v_{out})}$, respectively, is stored. Thus, for each point in the surface or Fourier plane a phase modulation pattern was generated to maximize the intensity at each targeted pixel:

$$\Phi_{SP}^{(x_{out},y_{out})}(u_{in}, v_{in}) = \arg\left(\sum_{j_x=1}^{25}\sum_{j_y=1}^{25}\exp\left(i\phi_{scan}^{j_x,j_y,p_{opt,SP}^{j_x,j_y,(x_{out},y_{out})}}(u_{in}, v_{in})\right)\right) \tag{1}$$

$$\Phi_{FP}^{(u_{out},v_{out})}(u_{in}, v_{in}) = \arg\left(\sum_{j_x=1}^{25}\sum_{j_y=1}^{25}\exp\left(i\phi_{scan}^{j_x,j_y,p_{opt,FP}^{j_x,j_y,(u_{out},v_{out})}}(u_{in}, v_{in})\right)\right) \tag{2}$$

This computation was then completed on the GPU for all the output foci in both SP and FP. This computational step took approximately 5 minutes for all 900 phase masks. Thus the total time for a single internal reference measurement was approximately 14 minutes. In principle it can completed faster by taking the inverse Fourier transform of the measured transmission matrix $p_{opt,SP}^{j_x,j_y,(x_{out},y_{out})}$ or $p_{opt,FP}^{j_x,j_y,(x_{out},y_{out})}$.

Serial internal reference beam calibrations were performed by calibrating multiple times with the internal reference beam focused on different regions of the input facet of the fiber, modulating the $c$ and $d$ parameters. The process is illustrated in **Fig 1B** whereby $B_{scan}^{j_x,j_y,p}$ raster scans the input core and $B_{ref}$ is indexed by $j_{ref}$ (where $1 < j_{ref} < N_{ref}$ and $N_{ref}$ is the total number of reference beams). In this case, the measurement is governed by 4 indices $j_x, j_y, j_{ref}, p$ resulting in a set of optimized phase shifts $p_{opt,SP}^{j_x,j_y,j_{ref},(x_{out},y_{out})}$, and phase masks $\Phi_{SP}^{(x_{out},y_{out},j_{ref})}(u_{in}, v_{in})$. Then, the index $j_{ref_{opt}}$ and $\Phi_{SP}^{(x_{out},y_{out},j_{ref_{opt}})}(u_{in}, v_{in})$ giving the highest intensity at $(x_{out}, y_{out})$ was selected and $\Phi_{SP}^{(x_{out},y_{out},j_{ref_{opt}})}(u_{in}, v_{in})$ was stored. Serialized reference calibrations were performed exclusively in the surface plane.

## Results and discussion

Firstly, the impact of the choice of internal reference beam was analysed for calibrations at the output facet's surface plane. Four calibrations were performed with the reference beam focussed on a different sub- region of the input facet of the fiber. The reference beams $B_{ref}^i$ (imaged on CCD1) were evenly shifted (by approximately 6 $\mu m$) along the radial direction from the very edge to the centre of the fiber core (left to right in the figure), as shown in **Fig 2A**. CCD1 was allowed to saturate so that the reflected speckle patterns were clearly visible. Each coherent reference beam $B_{ref}^i$ was scrambled by the turbid resulting in four different speckle patterns on the output facet on CCD2. Henceforth, they are referred to as $f_{SP}(B_{ref}^i)$, shown in **Fig 2B**. The fiber was ~ 4 cm long and parallel to the optical axis of the focussing objective. This was chosen so that, due to the short length and orientation of the fiber, each transmitted reference beam $f_{SP}(B_{ref}^i)$ would be well differentiated from each other in terms of its amplitude profile, thus, the dependence of the calibration quality on the amplitude pattern and as a consequence on the $i$th input position could be assessed. Indeed, it has been reported that for a short straight fiber, when a central mode is excited, the transmitted intensity remains largely confined to the centre [31]. In contrast, when an edge mode is excited at the input, the light is totally scrambled over a shorter length. This is also clear from the amplitude distributions of $f_{SP}(B_{ref}^i)$ in **Fig 2B**. Notably, the overall intensity transmitted by the edge aligned

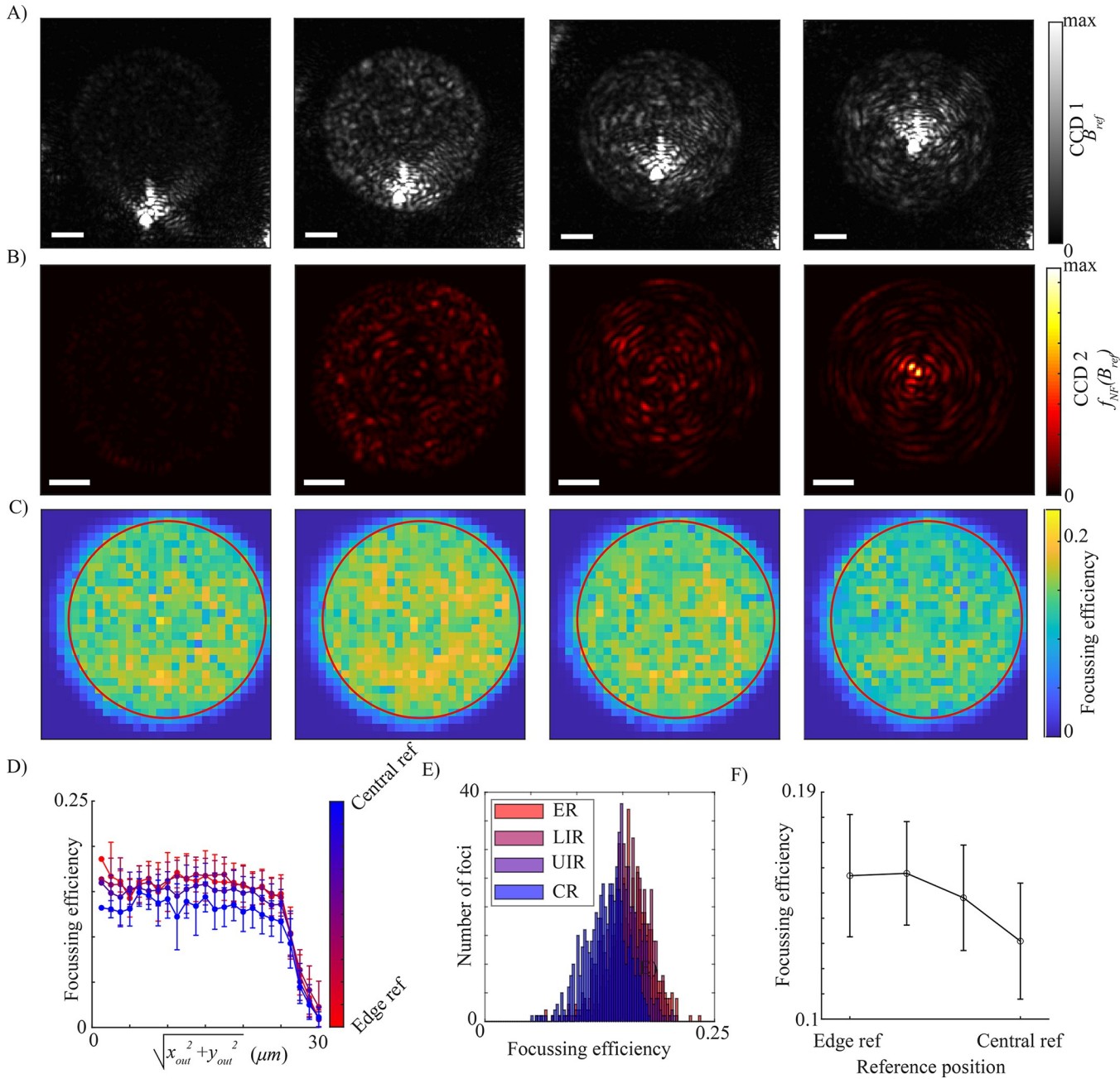

**Fig 2.** Analysis of choice of internal reference beam for surface plane calibrations A) Four internal reference beams on the input facet $B^i_{ref}$, imaged on CCD1. The position is shifted from the edge to centre of the core to centre (left to right) (scale bar 10 μm). B) The transmitted internal reference beams $f_{SP}(B^i_{ref})$ imaged on CCD2 (scale bar 10 μm). C) Focussing efficiency across the fiber core for each calibration). D) Dependence of focussing efficiency on radial position of fiber output as the reference is shifted from edge to centre (red to blue). E) Histograms of focussing efficiency (for foci within the fiber core) for measurements based on edge reference–ER, lower intermediate reference—LRI, upper intermediate reference–URI and central reference—CR. F) The fall in average focusing efficiency between edge and centre reference beam based calibrations.

reference is much lower than the other 3 examples, which could potentially be accounted for as an excitation at the boundary between core and cladding. Ultimately, all reference beams resulted in the ability to focus light on the output surface of the fiber but a different quality focusing across the output facet was observed.

The quality of the calibration was therefore analysed by estimating the focussing efficiency (FE) across the output facet, defined as $FE = \frac{I_{foci} - \varepsilon_{foci}}{I_{total} - \varepsilon_{total}}$ where $I_{foci}$ is the intensity within the SLM-generated foci, $I_{total}$ is the total intensity transmitted by the fiber and $\varepsilon_{foci}$ and $\varepsilon_{total}$ are the total readout electrical noise of the CMOS camera in the foci and in the entire facet, respectively. In each calibration, a 30 by 30 array of foci was targeted at the output facet. In each calibration, a 30 by 30 array of foci was targeted at the output facet. The focussing efficiency on the ($x_{out}$, $y_{out}$) plane, obtained by raster-scanning the output beam, is shown in **Fig 2C**. For a better quantification and comparison between the four investigated reference positions, **Fig 2D** and **2E** show the dependence of the focussing efficiency on radial distance from the centre of the core and histograms of focussing efficiency (for points within the core).

The calibration quality shows a clear dependence on the choice of reference beam. Notably, the central reference generates considerably lower focussing efficiency than the other 3 examples (the average over the entire core is approximately 1.15 times less (see **Fig 2F**), most strikingly at the centre of the core). This can be accounted for by considering the intensity difference between $f_{SP}(B^i_{ref})$ and $f_{SP}(B^{j_x,j_y,P}_{scan})$ for each input pair ($j_x$, $j_y$). The intensity of an interference pattern between two plane waves $I_1$ and $I_2$ is given by the well known equation [32]

$$I_{interfere} = I_1 + I_2 + 2\sqrt{I_1 I_2}\cos(\varphi_1 - \varphi_2) \tag{3}$$

Thus, the strongest phase dependency will occur when $f_{SP}(B^i_{ref})$ and $f_{SP}(B^{j_x,j_y,P}_{scan})$ are approximately equal in intensity. In the case of the central reference, the intensity of $f_{SP}(B^i_{ref})$ is unevenly distributed over the core with a greater proportion of intensity concentrated in the centre.

Hence, most modes will poorly interfere in this region of the fiber and weaker calibrations are produced. Therefore, in this case the optimal choice for an internal reference beam is toward the edge of the fiber due to the uniformity of the speckle pattern. Indeed, even the very weak reference beam generated at the core cladding interface results in a greater focussing efficiency than the central reference beam.

The uniformity of the resultant focussing efficiency map appears to be directly related to the the intensity profile of the reference beam. Going further, we have investigated the impact of fiber length and have performed calibrations akin to that in **Fig 2** for a fiber approximately 60 cm long. Results are reported in **S1 Fig in S1 File**. As in the previous dataset, the reference beam was evenly shifted (by approximately 6 μm) along the radial direction from the very edge to the centre of the fiber core (left to right in **S1 Fig in S1 File**). The resultant output speckle $f_{SP}(B^i_{ref})$ are shown in **S1B Fig in S1 File**. Although there is still clearly some preservation, the resultant intensity distributions are much more uniform than that for 4 cm fiber. This is in turn reflected in the focussing efficiency maps which are also more uniform for the central reference beam (comparing the far right panels of **Fig 2C** and **S1C Fig in S1 File**). For both the 4 cm and 60 cm fibers, an edge reference results in higher average focussing efficiency across the fiber core, however this is significantly more pronounced for the 4 cm fiber. Overall, the average focussing efficiency over the core was slightly higher for the short fiber for all reference beams and the average focussing efficiency was much less sensitive to the choice of reference beam for the longer fiber. As well as assessing the role of fiber length and the amplitude profile of $f_{SP}(B^i_{ref})$, in **S2 Fig in S1 File**, we also asses the role of the phase profile of $f_{SP}(B^i_{ref})$. The phase profile was measured using an external reference and the phase unwrapping technique described in [3]. The phase profile is shown in **S2A Fig in S1 File** and the resultant focussing efficiency map is shown in **S2B Fig in S1 File**. No trace of the phase profile can be found in the

resultant focussing efficiency map, however it is noteworthy that the output foci are no longer phase aligned (as they would be for a uniform phase profile external reference).

A single internal reference-based calibration is compared with an external reference based calibration in **S3 Fig in S1 File**. The optical setup was modified as shown in **S3 Fig in S1 File**, so that after the first HWP the laser was split by 50:50 beam splitter BS1, its polarisation was rotated again by HWP2 to match with that of the fiber transmission. The two beams were re-joined by BS3 and a linear polariser was placed in front of the screen of CCD2. Panel B shows the focusing efficiency two datasets generated with an external reference (left) and an internal reference (right) and panel C shows the histogram of focussing effiencies. The advantage of the internal reference is evident both in terms of the evenness of the calibration across the fiber core and maximum intensity attainable. Between the two datasets, the average focusing efficiency increased from 6% to 15%. Although external reference based calibrations have been demonstrated in the literature [33] to achieve significantly higher focussing efficiency, they require extensive spatial filtering, beam shaping and feedback loops to achieve the high level of interferomic stability required for this process. The data in **S3 Fig in S1 File** shows that, on the other hand, internal reference based calibration offers a neat solution by an inherently stable interferometric system.

We then performed the calibration in the Fourier plane denoted ($u_{out}$,$v_{out}$). Our previous publication has characterised Fourier plane calibrations based on an external reference beam [23] and far-field imaging through MMFs has recently gained attention as a method to image objects distal from the fiber tip [24]. In the Fourier plane the resolution limit ($\theta_{width}$) of the focussed spots is given by $sin(\theta_{width}) = \frac{1.22\lambda}{d} = \frac{1.22 \times 633\ nm}{50\ \mu m}$. As in the surface plane experiment, the calibration was performed with the reference beam in 4 different positions at the input, displayed in **Fig 3A**. **Fig 3B** shows the fourier plane transmission by the fiber $f_{FP}(B^i_{ref})$ for each of the reference beams, imaged on CCD3. As in the surface plane, the central reference better preserves the structuring of the input light and in this case more intensity is concentrated at the center of the fiber. The spatial dependency of the focussing efficiency of each Fourier plane foci is shown in **Fig 3C** and **3D**, and by a histogram in **Fig 3E**. Previous publications where an external reference beam was used, have demonstrated that typically the intensity of the Fourier plane foci falls for higher angles still within the NA of the fiber [23, 24]. This also appears to be the case for internal reference beam calibrations, however the data in **Fig 3** shows that the picture is further complicated by the interferometric compatibility of the two beams. In the case of the central reference beam, weaker foci are generated at the centre and highest intensity foci are generated at approximately half the NA, which can be accounted for by considering the profile of $f_{FP}(B^i_{ref})$ and Eq 3.

For edge reference beams, the stronger foci may be generated at the half of the fibers numerical aperture, as in the external reference experiments. However, within the intensity distribution, a second shoulder peak at higher angles is observed. At higher angles, the profile of the reference $f_{FP}(B^i_{ref})$ is also uneven with an increase in intensity favoured at the NA boundary which could tentatively be claimed as accounting for this behaviour.

The above results demonstrate that in both the surface and Fourier planes, the calibration quality is highly dependent on the intensity and amplitude profile of the transmitted reference beam. Therefore, selecting a reference beam with uniform speckle pattern at the output is of the upmost importance. The histograms of the focussing efficiency also exhibit shoulder peaks at lower values corresponding to "*blind spots*" where the reference beam has a phase singularity. An example of this is shown in **Fig 4A**, where a characteristic focussed spot and a blind spot are shown side by side. To solve this, multiple calibrations with different internal reference beams can be employed, as illustrated in **Fig 1B(ii)** and described in the recent

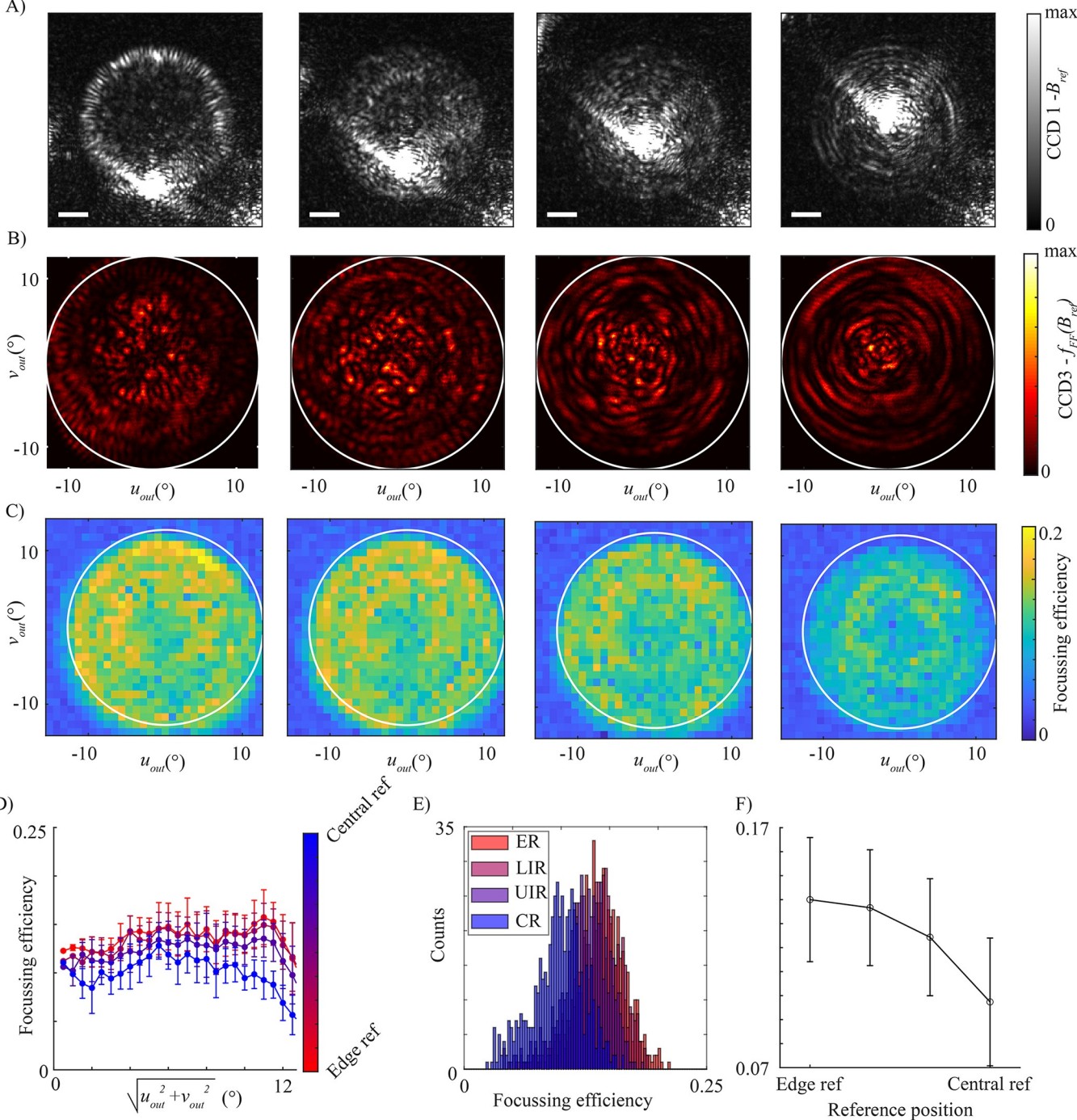

**Fig 3.** Analysis of internal choice of internal reference beam for fourier plane calibrations A) Four internal reference beams at the input facet $B_{ref}^i$ (scale bar 10 µm). B) The transmitted internal reference beams $f_{FP}(B_{ref}^i)$ imaged on CCD3, the NA limit of the fiber is shown as a white circle. C) Focussing efficiency across the fiber NA for each calibration D) Dependence of focussing efficiency on output angle of fiber facet. E) Histograms of focussing efficiency (for foci within the fiber NA) for measurements based on edge reference–ER, lower intermediate reference—LRI, upper intermediate reference–URI and central reference—CR. F) The fall in average focusing efficiency between edge and centre reference based calibrations.

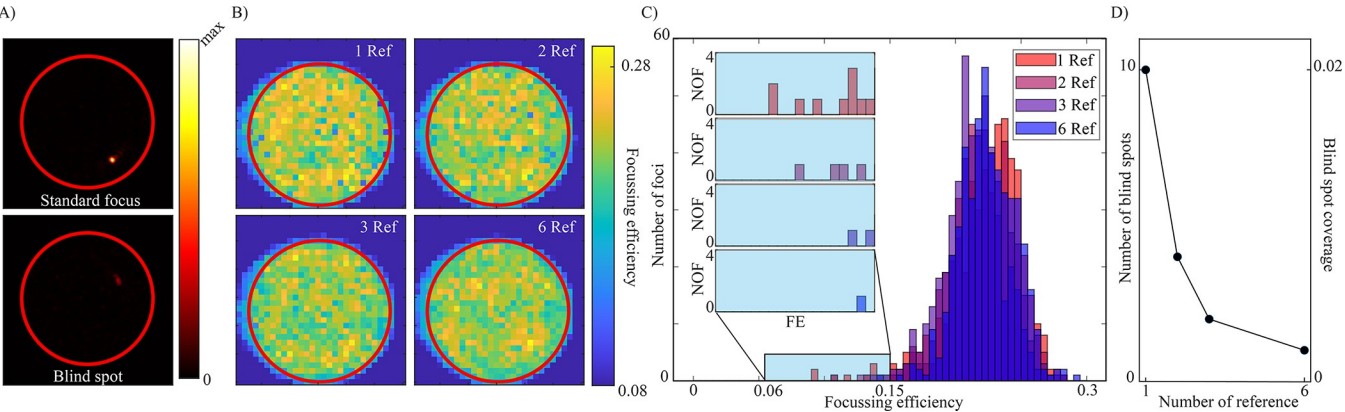

**Fig 4. Analysis of calibrations based on serialised internal reference beams.** A) Characteristic example of a focussed spot on the fiber surface (top) and a blind spot (bottom) B) Focussing efficiency for four serialised internal reference based calibrations based on single reference (top left), two references (top right), three references (bottom left) and six references (bottom right) C) Histograms of the focussing efficiency for the four measurements, the inset shows the highlighted region which were classified as blind spots. D) The average number of blind spots in each calibration (left y-axis) and the blind spot coverage ratio (no of blind spots/no of focussed spots) (right y-axis), based on 3 experiments. E) Multi-spot focussing through a MMF based on single (top) and multiple (bottom) internal reference calibrations.

publication [30]. Focusing efficiency maps for four surface plane calibrations are shown in **Fig 4B** where between 1 and 6 reference beams were used. The reference beams $B_{ref}^i$ were evenly displaced along the radius of the core at the input facet and a calibration for each reference was performed. For each position, the calibration generating the strongest foci was saved, while the others are discarded. The calibration took approximately 30 minutes per reference beam. From the colormaps in panel **Fig 4B** it is clear that in all cases focussing efficiency is relatively stable across the core and independent of radial distance from the center of the core. However, the single reference-based measurement exhibits a number of blind spots that are evident at the tail of the histograms in **Fig 4C** (highlighted by the blue rectangle) (they correspond to blue pixels in the FE maps in **Fig 4B**). Based on a fitted normal distribution over the histograms, we classified points with focussing efficiency below 15% as blind spots. A final figure of merit for this experiment is shown in **Fig 4D**, where the number of blind spots for each measurement is shown The average number of blind spots falls from 8.67±4.16 with a single reference calibration to 0.33±0.58 for the 6 reference beam based calibration (based on $n = 3$ experiments). Thus, it can be assessed that at the expense of higher computational time, internal reference based calibration can be applied to improve the calibration quality. The final panel of **Fig 4** shows how two foci may be generated simultaneously at two output points with both a single and multiple reference calibration following the formula:

$$\Phi_{SP}^{(x_1,y_1),(x_2,y_2)} = arg(exp(i\Phi_{SP}^{(x_1,y_1)}) + exp(i\Phi_{SP}^{(x_2,y_2)}))$$

With the focusing efficiency shared between them. As this computation discards the amplitude information, the intensity distribution is not impacted by the number of reference beams used to calibrate.

## Conclusions

Consequentially, internal reference beams appear to be an attractive basis for controlling transmission through optical fibers. Chosing an internal reference beam with an even amplitude distribution in the calibration plane leads to strong interformetric contrast between $f_{SP}(B_{ref}^i)$ and $f_{SP}(B_{scan}^{ix,iy})$ (or $f_{FP}(B_{ref}^i)$ and $f_{FP}(B_{scan}^{ix,iy})$). Although isolated blind spots appear in the

calibration, this can be overcome (at the expense of computational time) by using a series of internal reference beams, which simultaneously increases focussing efficiency. Overall, when compared with external reference beams, the common optical path of an internal reference-based system easily achieves a high interferometric stability without the requirement of a complex optical or machine learning setup. Overall, our study demonstrates the utility of internal reference-based systems, which is inherently reconfigurable, we envisage this will be crucial in allowing the technology to reach its full potential for hyper-spectral imaging at the tip of a hair thin optical fiber.

## Supporting information

**S1 File.**
(DOCX)

## Author Contributions

**Conceptualization:** Liam Collard, Filippo Pisano, Di Zheng, Massimo De Vittorio, Ferruccio Pisanello.

**Data curation:** Liam Collard, Linda Piscopo.

**Formal analysis:** Liam Collard, Linda Piscopo, Ferruccio Pisanello.

**Funding acquisition:** Massimo De Vittorio, Ferruccio Pisanello.

**Investigation:** Liam Collard, Linda Piscopo, Filippo Pisano, Di Zheng, Massimo De Vittorio, Ferruccio Pisanello.

**Methodology:** Liam Collard, Filippo Pisano, Di Zheng, Massimo De Vittorio, Ferruccio Pisanello.

**Project administration:** Ferruccio Pisanello.

**Resources:** Ferruccio Pisanello.

**Software:** Liam Collard, Ferruccio Pisanello.

**Supervision:** Massimo De Vittorio, Ferruccio Pisanello.

**Validation:** Liam Collard, Filippo Pisano, Di Zheng, Massimo De Vittorio, Ferruccio Pisanello.

**Visualization:** Liam Collard, Linda Piscopo, Ferruccio Pisanello.

**Writing – original draft:** Liam Collard, Ferruccio Pisanello.

**Writing – review & editing:** Liam Collard, Linda Piscopo, Filippo Pisano, Di Zheng, Massimo De Vittorio, Ferruccio Pisanello.

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
