## [Decision Letter · Decision Letter 0]

1 Mar 2023

PONE-D-23-02315Determining the optimal reference beam to measure the transmission matrix of a multimode optical fiberPLOS ONE

Dear Dr. Collard,

Thank you for submitting your manuscript to PLOS ONE. After careful consideration, we feel that it has merit but does not fully meet PLOS ONE’s publication criteria as it currently stands. Therefore, we invite you to submit a revised version of the manuscript that addresses the points raised during the review process

We look forward to receiving your revised manuscript.

Kind regards,

Yuan-Fong Chou Chau

Academic Editor

PLOS ONE

Journal Requirements:

"L.C., D.Z., M.D.V., and Fe.P. acknowledge European Union's Horizon 2020 Research and Innovation Program under Grant Agreement No. 828972. Fi.P., and Fe.P. acknowledge European Research Council under the European Union's Horizon 2020 Research and Innovation Program under Grant Agreement No. 677683. Fi.P., M.D.V., and Fe.P. acknowledge European Union's Horizon 2020 Research and Innovation Program under Grant Agreement No 101016787. M.D.V. acknowledges European Research Council under the European Union's Horizon 2020 Research and Innovation Program under Grant Agreement No. 692943. M.D.V. acknowledges U.S. National Institutes of Health (Grant No. U01NS094190). M.D.V., and Fe.P. acknowledge U.S. National Institutes of Health (Grant No. 1UF1NS108177-01)." 

"I have read the journal’s policy and the authors of this manuscript have the following competing interests: M.D.V. and F.Pisanello are founders and hold private equity in OptogeniX srl, a company that develops, produces and sells technologies to deliver light into the brain. This does not alter our adherence to PLOS ONE policies on sharing data and materials. OptogeniX did not fund the research described in this work. M.D.V.: OptogeniX srl (I). F.P.: OptogeniX srl (I)."

Please confirm that this does not alter your adherence to all PLOS ONE policies on sharing data and materials, by including the following statement: ""This does not alter our adherence to  PLOS ONE policies on sharing data and materials.” (as detailed online in our guide for authors http://journals.plos.org/plosone/s/competing-interests).  If there are restrictions on sharing of data and/or materials, please state these. 

Please note that we cannot proceed with consideration of your article until this information has been declared. 

Reviewers' comments:

Reviewer's Responses to Questions

**Comments to the Author**

1. Is the manuscript technically sound, and do the data support the conclusions?

Reviewer #1: Yes

Reviewer #2: Partly

Reviewer #3: Yes

Reviewer #4: Yes

2. Has the statistical analysis been performed appropriately and rigorously? 

Reviewer #1: Yes

Reviewer #2: No

Reviewer #3: Yes

Reviewer #4: Yes

3. Have the authors made all data underlying the findings in their manuscript fully available?

Reviewer #1: Yes

Reviewer #2: Yes

Reviewer #3: Yes

Reviewer #4: Yes

4. Is the manuscript presented in an intelligible fashion and written in standard English?

Reviewer #1: Yes

Reviewer #2: Yes

Reviewer #3: Yes

Reviewer #4: Yes

5. Review Comments to the Author

Reviewer #1: In this paper, a reference beam optimization scheme for measuring transmission matrix of multimode fiber based on the internal reference beam is proposed through experiments and analysis. By changing the coupling position and number of the internal reference beam and analyzing the speckle distribution and Fourier transform spectral intensity of the output facet of optical fiber, the author summarizes the implementation scheme of the inner reference beam. This manuscript is written logically and clearly shown the purpose. It may be accepted for publishing in PONE after a minor revision, but there are still several problems.

1. The author said “The numerical aperture of MO1 was significantly higher than that of the MMF. This was done so that the input modes of the fiber could be fully sampled”. Please explain that in detail.

2. It seems that the choice of reference beam can be determined from the analysis of the output face of the fiber. Why do we need to do Fourier analysis?

3. Obviously, in order to record the speckle field completely, the speckle field of the reference beam must be uniform enough. So, are there other factors other than the appropriate focusing position described in the article? Such as numerical aperture of MO, quality of the reference beam, diameter of fiber, etc.

4. Can the blind spots of the inner reference beam be eliminated completely?

5. In Figures 2 and 3, the scale bar is not indicated. In Figure 3(e), the number “35” is blocked.

6. In Figure 3, what do the �u and �v refer to?

Reviewer #2: In the manuscript tilted “Determining the optimal reference beam to measure the transmission matrix of a multimode optical fiber”, the authors present the merits of using internal reference method to improve focusing through a multimode fiber. The internal reference method is a great idea in my opinion, but the work presented by the author needs to ‘dig a bit deeper’.

The best way to generate a single focus using wavefront shaping method is the step sequential method introduced at the dawn of the technique. The transmission matrix method is the most accurate but it requires determining both amplitude and phase of the transmitted signal. The most accurate method to obtained the TM uses some sort of external/stable reference beam (holographic method) or uses phase retrieval techniques (a bit more computationally intensive). All the above methods can generate/transfer images through multimode fibers.

Here the authors only demonstrate their work for a single focus. I believe Collard et al. will have to put their work more in context to justify this conclusion “Overall, our study demonstrates the utility of internal reference-based systems, which is inherently reconfigurable, we envisage this will be crucial in allowing the technology to reach its full potential for hyper-spectral imaging at the tip of a hair thin optical fiber.”

See below for further questions and concerns:

1. The authors can do a little bit more of digging in the literature. There is quite a comprehensive body of work for wavefront shaping in multimode fiber both with external reference, internal reference, and reference-less (phase retrieval). How is this method compare with the other well established endoscopic imaging method?

2. The authors should revised their references. For example Reference [24] is not a reference less method. They have an external reference going through a second fiber. I do not believe they claim AI. The references for [25] and [26] aren’t accurate either. I believe the numbers have been shifted down. Please revise.

3. The length of the fiber (4 cm) used in this experiment is of concern. As far as I know, a lot of the single fiber endoscopic method in the literature have much longer fibers, some with much larger core (more modes). How will this method presented here, work when the fiber is 10 or 20 times longer? This isn’t obvious to me; in a longer fiber the light are even more scrambled. Further, bends, temperature, vibrations, etc. will also affect both the signal and reference. How is this accounted for in this study?

4. Further, the most accurate holographic techniques use a spatially filtered external reference beam: the phase and intensity of the beam well controlled and known. The idea of an internal reference is indeed fascinating, but the internal reference will also suffer the same fate as the signal beam. Can the author discuss how is the amplitude and phase of the internal reference determined? In equation (3) in the manuscript, the intensity of the reference beam is measured accurately, but how is the phase determined?

5. How is the work presented in this manuscript improve from what has been done in reference [26]? What is novel? Different?

6. There is a discussion on a single focus. Wavefront shaping have significantly moved on to multiple foci and image transfer. How will the changes in a multiple internal reference affect a multiple foci problem? An image?

I hope the authors will find my reviews helpful and hopefully improve the manuscript

Reviewer #3: In this manuscript, the authors investigated the conditions that affect the optimization of reference beams for multi-mode optical fiber’s transmission matrix measurements based on the common-path interferometric method. The authors defined the focusing efficiency and used this as a metric to experimentally investigated the optimal parameters for determining the internal reference beam. Related experimental data can support their claims. This work helps establish the metrics and method to optimize the reference beam as well as investigate the optimal experimental scheme to this end. Since wavefront-shaping-based optical fiber transmission matrix calibrations play a central role in fiber imaging, this work can benefit related fiber-based biomedical imaging research to a large extent. Overall, I would like to recommend the publication of this work. However, The writing of this manuscript has some minor issues. The authors need to address the following minor issues in their revised manuscript.

1. In the authors’ definition of focusing efficiency, it is not quite clear how the authors define “dark currents”? The physics meaning of the “dark currents” should be clarified.

2. In Figure 1, there are supposed to be four focal planes. However, in the experimental system drawing, only three focal planes are illustrated.

3. All the figures in this manuscript use upper-case letters. However, when the authors cited the figures in their main text, they used lowercase letters, which should be corrected.

4. The authors should add the length of the scale bar in Figure 2 and Figure 3.

5. Figure 2 d is confusing. Curves with different colors indicate focusing efficiency correlated with different spatial locations. Then, what is the meaning of the horizontal axis?

Reviewer #4: In the article the authors investigate to achieve desired amplitude distributions at the tip of a multimode

fiber (MMF) by using pre-shaped light. The model is extremely simple: they use two sawtooth gratings with some phase shift between together. It allows to control a little bit the intensity and the output of short 4-cm long MMF. The work has a lot of drawbacks that should be fixed before publication:

1. The title is confusing. In fact the authors did not present a measured transmission matrix as is has been done in other works like [1-3].

2. In the abstract the authors refer to optically turbid waveguide, but a standard MMF that is absolutely transparent for light was used in the experiments. What did it really mean?

3. SP and FP planes have not denoted in the Fig.1 but mentioned in the text.

4. Why the authors use such simply expression and do not mention other more improved methods like [4] for scanning?

5. The phase shift 'p' is unclear and there is no equation how exactly it affects to the sawtooth gratings.

6. Which array of points was used, 25x25 or 30x30? Both of them can be found in the text.

7. The used MMF was only 4-cm long. Are there any applications where it would be in demand? For me such length is impractical.

8. The first paragraph on page 9 describes the figure from supplementary that is really strange... It would be better to modify Fig1 to show external reverence beam and add one more figure with the characteristics of the focal point or prepare self-consistent supplementary.

Finally, I believe that a major revision required.

References:

1. https://www.mdpi.com/2076-3417/9/1/195#

2. https://doi.org/10.1364/OL.41.005580

3. https://doi.org/10.1364/OE.389133

4. https://doi.org/10.1364/OE.15.001913

6. PLOS authors have the option to publish the peer review history of their article (what does this mean?). If published, this will include your full peer review and any attached files.

Reviewer #1: **Yes: **Jianglei Di

Reviewer #2: **Yes: **Dr. Moussa N'Gom

Reviewer #3: No

Reviewer #4: No

---

## [Author Response · Author response to Decision Letter 0]

15 Jun 2023

Dear Professor Chou Chau,

Thank you for your efforts in reviewing our manuscript "Determining the optimal reference beam to measure the transmission matrix of a multimode optical fiber PONE-D-23-02315” Enclosed is our revised manuscript, supporting information and response to the reviewers. 

We are grateful to the reviewers for their thorough reading of our manuscript and their detailed comments. In our response letter we have provided a detailed point by point response to the reviewers’ questions. In particular, a new figure has been added to the supporting information addressing the points raised by several reviewers about the dependence of our method on fiber length. As well as this, new measurements have been performed where the phase of the internal reference beam was measured and compared with the resultant focusing efficiency map.

As well as this, the introduction, discussion sections have been modified to provide a better comparison with the state of the art following the comments from all reviewers.

Following the comments from reviewer 4, we would also like to change the title of work to:

“Optimizing the internal phase reference to shape the output of a multimode optical fiber”

On behalf of all authors,

Yours sincerely,

Dr Liam Collard

---

## [Decision Letter · Decision Letter 1]

31 Jul 2023

PONE-D-23-02315R1Optimizing the internal phase reference to shape the output of a multimode optical fiberPLOS ONE

Dear Dr. Collard,

Thank you for submitting your manuscript to PLOS ONE. After careful consideration, we feel that it has merit but does not fully meet PLOS ONE’s publication criteria as it currently stands. Therefore, we invite you to submit a revised version of the manuscript that addresses the points raised during the review process.

We look forward to receiving your revised manuscript.

Kind regards,

Yuan-Fong Chou Chau

Academic Editor

PLOS ONE

Journal Requirements:

Reviewers' comments:

Reviewer's Responses to Questions

**Comments to the Author**

1. If the authors have adequately addressed your comments raised in a previous round of review and you feel that this manuscript is now acceptable for publication, you may indicate that here to bypass the “Comments to the Author” section, enter your conflict of interest statement in the “Confidential to Editor” section, and submit your "Accept" recommendation.

Reviewer #5: (No Response)

2. Is the manuscript technically sound, and do the data support the conclusions?

Reviewer #5: Yes

3. Has the statistical analysis been performed appropriately and rigorously? 

Reviewer #5: Yes

4. Have the authors made all data underlying the findings in their manuscript fully available?

Reviewer #5: Yes

5. Is the manuscript presented in an intelligible fashion and written in standard English?

Reviewer #5: No

6. Review Comments to the Author

Reviewer #5: In this manuscript, the authors perform assessments for both surface and Fourier plane calibrations. In the surface calibration, they focus the light on the distal facet, and in the Fourier plane calibration, they collimate the light into a low-divergence beam emerging from the distal facet. To optimize the system's relative focusing efficiency, they employ both single and serialized internal reference beams. This work may deserve publication in my opinion.

However, several revisions, including some major ones, still need to be addressed.

1) The authors should carefully check their notations as there are several unacceptable notation errors in the manuscript. Some are listed below:

1a) On page 26, it should be "(u_in, v_in)" instead of "(u_in, y_in)."

1b) In Figure 2D, the x-axis should be labeled as "\\sqrt{x_out^2, y_out^2)}" instead of "\\sqrt{x_out^2, v_out^2)."

1c) On page 34, the correct notation for the calibration in the Fourier plane is "(u_out, v_out)" rather than "(u_out_v_out)."

2) Figure S3 has been incorrectly labeled as Figure S1 in the Supporting Information. To rectify this error, the authors must ensure that the correct label, "Figure S3," is used for the respective figure in the Supporting Information section. Moreover, it is essential to update the caption in Figure S3 to accurately correspond to the content of Figure S3.

3) The meaning of "S1B" on page 31 and "S2B" on page 32 is unclear. Should they be Figure S1B and Figure S2B?

4) Following QUESTION R3.2, the focal planes for (u_in, v_in) and (u_out, v_out) in Figure 1A are challenging to discern. To address this issue, the authors should consider modifying the colors used for these focal planes to enhance visibility and clarity.

5) Following QUESTION R3.3, the issue has not been fully addressed yet. For example, Figure 2b on page 29 and Figure 2c on page 32 still require corrections. Additionally, there are inconsistencies in the citation format within the main text, where certain figure references are bolded while others are not. Moreover, the authors sometimes use "Figure S1" and other times "figure S1."

6) The authors should include a legend to distinguish between the two different colors in the histograms for both Figure 2E and Figure 3E.

7) In addition to mentioning the information in the caption, the authors should also label the number of references in Figure 4B.

8) The "um" in the x-axis of Figure S1D should be corrected to "μm."

9) The caption of Figure 2 requires revision to include the scale bar information for Figure 2A, which is currently missing. It's important to note that the mentioned 10 m scale bar is not present in Figure 2C, despite being referenced in the caption.

10) The colorbar label in Figure 3C, Figure S2A, Figure S2B, and Figure S3B should be horizontally flipped to make them consistently.

11) The authors should be mindful of their use of abbreviations to prevent redundancy. For example, on page 28, they introduce the abbreviation "SP" for "surface plane." However, on page 29, they opt to abbreviate "surface plane" once more instead of using the established abbreviation "SP."

12) There are punctuation errors that need to be addressed. In two instances (pages 26 and 28), the authors used ".." to end sentences, but it should be corrected to use ".", instead. Additionally, on page 31, the authors used ",." which should also be fixed to "." for proper punctuation.

7. PLOS authors have the option to publish the peer review history of their article (what does this mean?). If published, this will include your full peer review and any attached files.

Reviewer #5: No

---

## [Author Response · Author response to Decision Letter 1]

3 Aug 2023

Dear Professor Chou Chau,

Thank you for your continued efforts in reviewing our manuscript "Determining the optimal reference beam to measure the transmission matrix of a multimode optical fiber PONE-D-23-02315” Enclosed is our revised manuscript, supporting information and response to the reviewers. 

We are grateful to the new reviewer for their thorough reading of our manuscript and their detailed comments. In our response letter we have provided a detailed point by point response to the reviewers’ questions.

On behalf of all authors,

Yours sincerely,

Dr Liam Collard

---

## [Editor Report · Decision Letter 2]

4 Aug 2023

Optimizing the internal phase reference to shape the output of a multimode optical fiber

PONE-D-23-02315R2

Dear Dr. Collard,

We’re pleased to inform you that your manuscript has been judged scientifically suitable for publication and will be formally accepted for publication once it meets all outstanding technical requirements.

Kind regards,

Yuan-Fong Chou Chau

Academic Editor

PLOS ONE
---

## [Editor Report · Acceptance letter]

31 Aug 2023

PONE-D-23-02315R2 

*Optimizing the internal phase reference to shape the output of a multimode optical fiber*

Dear Dr. Collard:

I'm pleased to inform you that your manuscript has been deemed suitable for publication in PLOS ONE. Congratulations! Your manuscript is now with our production department. 

Kind regards, 

on behalf of

Dr. Yuan-Fong Chou Chau 

Academic Editor

PLOS ONE